# The Antioxidant, Anti-Platelet and Anti-Coagulant Properties of Phenolic Compounds, Associated with Modulation of Hemostasis and Cardiovascular Disease, and Their Possible Effect on COVID-19

**DOI:** 10.3390/nu14071390

**Published:** 2022-03-26

**Authors:** Beata Olas

**Affiliations:** Department of General Biochemistry, Faculty of Biology and Environmental Protection, University of Lodz, Pomorska 141/3, 90-236 Lodz, Poland; beata.olas@biol.uni.lodz.pl; Tel./Fax: +48-42-635-44-84

**Keywords:** COVID-19, cardiovascular disease, phenolic compounds

## Abstract

Patients affected by coronavirus disease 2019 (COVID-19) demonstrate a range of hemostasis dysfunctions, such as coagulation dysfunction and changes in blood platelet function, this being a major cause of death. These complications may also be associated with oxidative stress. Recently, various papers, including some reviews, have suggested that the use of dietary bioactive compounds, including phenolic compounds, may play a significant role in the treatment of COVID-19. However, while some phenolic compounds, such as curcumin, resveratrol, myricetin and scutellarian, have been found to have antiviral effects against COVID-19, recommendations regarding the use of such compounds to prevent or reduce the risk of CVDs during COVID-19 infection remain tentative. The present mini-review examines the antioxidant, anti-platelet and anticoagulant and antiviral activities of selected phenolic compounds and the possible implications for their use in treating CVDs associated with COVID-19. This review also examines whether these phenolic compounds can be promising agents in the modulation of hemostasis and CVDs during COVID-19. While their properties have been well documented in various in vitro and in vivo studies, particularly their positive role in the prophylaxis and treatment of CVDs, less is known regarding their prophylactic potential against CVDs during COVID-19, and no credible evidence exists for their efficiency in humans or animals. In such cases, no in vitro or in vivo studies are available. Therefore, it cannot be unequivocally stated whether treatment with these phenolic compounds offers benefits against CVDs in patients with COVID-19.

## 1. Introduction

Infection with SARS-CoV-2 is known to entail a range of sequelae. These can include various hemostasis dysfunctions, including coagulation dysfunction and changes in blood platelet function; this being a major cause of death. Such changes in hemostasis, for example, increased blood platelet activation, plays an important role in the development of CVDs. However, little is known of the participation of platelets in the pathogenesis of COVID-19, particularly the stages of blood platelet activation. Moreover, the elevated activity of the inflammatory response and the course of the clinical illness itself, as well as traditional risk factors, can increase the risk of thrombotic events [1,2,3,4,5,6,7,8,9,10,11,12,13,14,15,16,17,18,19,20]. More details about the modulation of hemostasis in COVID-19 and role of blood platelets are provided in a review by Ulanowska and Olas [21].

A diet rich in fruits and vegetables has a significant influence on hemostasis and CVDs. For example, various authors [22,23,24,25] have found that fruit and vegetable consumption lowers the risk of CVDs by influencing the activity of various elements of hemostasis, such as coagulation proteins, and by reducing oxidative stress in the cardiovascular system and inhibiting blood platelet activation.

While the drugs used to treat COVID-19 are intended only for symptomatic treatment, various papers have suggested that dietary bioactive compounds, including phenolic compounds, may also play a role in COVID-19 therapeutics. However, although studies suggest that phenolic compounds may have the potential to treat COVID-19, these papers found them to have little antiviral potential [26,27,28,29,30,31,32,33,34].

A number of recent review papers, and some experimental works, suggest that the use of dietary bioactive compounds, including phenolic compounds may play an important role in COVID-19 therapeutics [35,36,37,38]. However, it is important to emphasize that these findings are tentative, and that the authors only suggest that their antioxidant, anti-platelet and anticoagulant activities may serve as the basis of alternative therapeutic approaches for preventing, or reducing, the risk of CVDs in patients with COVID-19. For example, a review by Giordo et al. [30] found resveratrol to have therapeutic potential in CVDs associated with COVID-19; however, this recommendation is only based on results from in vitro experiments.

The present mini-review describes the antioxidant, anti-platelet and anticoagulant properties of selected phenolic compounds and their possible effects of their administration on CVDs associated with COVID-19; however, the paper only examines selected phenolic compounds which have previously been found to demonstrate antiviral activities. In addition, it examines whether these phenolic compounds exert a beneficial effect on hemostasis and CVDs in patients with COVID-19. The potential of these compounds and their effectiveness in preventing modulation of hemostasis and CVDs are given in Table 1.

This review is based on studies identified in electronic databases, including PubMed, ScienceDirect, Web of Knowledge, Google Scholar, and SCOPUS. The last search was run on 10 December 2021. The following terms were used (publication date—10 years): “COVID-19” (the number of articles: 201,341), “phenolic compounds” (the number of articles: 45,342), or “phenolics” (the number of articles: 238,876), or “phenolic compound with antiviral activity” (the number of articles: 987), “oxidative stress” (the number of articles: 156,765), or “antioxidant” (the number of articles: 445,654), and “hemostasis” (the number of articles: 98,321), or “coagulation” (the number of articles: 300,378), or “blood platelets” (the number of articles: 78,765), or “thrombosis” (the number of articles: 187,987), or “cardiovascular diseases” (the number of articles: 678,654).

## 2. Antioxidant, Anti-Platelet and Anticoagulant Potential of Phenolic Compounds

Phenolic compounds are not only found in medicinal plants, but also in the edible parts of the fruits and vegetables which form a key part of the human diet; they have also been identified in chocolate, dry legumes and cereals, and beverages such as tea, coffee and wine. While all phenolics possess one or more aromatic rings bearing one or more hydroxyl groups, they can be divided into several groups based on their chemical structure, such as flavonoids, phenolic acids, stilbenes, tannins, coumarins, lignans and curcuminoids; they can also occur in free forms or conjugated with sugars and acids. The compounds are not believed to be toxic; however, their Recommended Daily Intake remains unknown [22,23,24,25].

Due to their numerous and varied activities, these phenolic compounds have found a wide range of uses in different areas of life. As noted above, some have been found to offer health benefits against CVDs [22,23,24,25]; for example, flavonoid intake inversely correlated with mortality from CVDs in humans.

In addition, some phenolic compounds, including curcumin, resveratrol, myricetin and scutellarian appear to have antiviral effects against COVID-19. These antiviral effects are believed to act through various mechanisms, for example by reducing viral penetration, genome replication and gene expression. Some authors suggest that phenolic compounds such as quercetin, kaempferol, curcumin, naringenin and catechin may act against COVID-19 proteases. However, this antiviral potential has only been observed in in vitro models, for example for vero E6 cells [26,27,28,29,30,31,32,33,34]. Recently, Autodock docking analysis by Davella et al. [34] found two phenolic compounds from *Piper Nigrum* (black pepper), kadsurenin L and methysticin, to be highly susceptible to the COVID-19 major protease enzyme in silico.

### 2.1. Resveratrol

Among the several phenolic compounds which form an integral part of the human diet, one of the most interesting from a health perspective is resveratrol, a compound with not only antiviral potential, but also antioxidant activity. Resveratrol is a secondary metabolite produced by some species in response to harmful environmental factors, such as UV radiation, fungal, bacterial and viral infection, or mechanical damage to the plant. The name *resveratrol* probably arose as a blend of the name of the parent compound *resorcinol* and the plant from which it was derived, *Veratrum*. The -ol ending of course indicates the presence of hydroxyl groups [25,39].

While resveratrol was first isolated in 1939 from the root of the white oppressor (*Veratrum grandiflorum*), it has since been identified in praising charcoal (*Polygonum cuspidatum*), a plant used in traditional Chinese and Japanese medicine, which is now the primary source of commercial sourcing of pure resveratrol. Resveratrol is naturally found in about 72 plant species from inter alia the classes *Cyperaceae*, *Dipterocarpaceae*, *Gnetaceae* and *Vitaceae*. The best and most widely available natural source of resveratrol is the grape (*Vitis vinifera*), especially the darker varieties. However, resveratrol is also found in mulberry, black currant, blueberry, cowberry and American, strawberries, raspberries, apples and peanuts [25,39].

In vitro studies have demonstrated that resveratrol possesses antioxidant activity and decreases the level of biomarkers of oxidative stress associated with hemostasis. For example, preincubation of human blood platelets with 100 μM resveratrol was found to have a protective effect against thiol oxidation (including glutathione, cysteine and cysteinylglycine) induced by 100 μM peroxynitrite (ONOO^−^): a reactive oxygen species (ROS) produced in the reaction between the superoxide anion (O_2_^−^) and nitric oxide (NO). The level of thiols was measured by HPLC [40]. In addition, resveratrol has also been found to diminish tyrosine nitration (determined by competition C-ELISA and Western blot) and carbonylation in blood platelet proteins induced by ONOO^−^ [41,42]. Resveratrol (25 μg/mL) also countered the decrease of blood platelet thiols induced by platinum compounds such as cisplatin and selenium-cisplatin conjugate in vitro [43]. It also reduced ROS production in resting platelets and platelets activated by various agonists, including thrombin [44] and demonstrated protective effects against oxidative and nitrative protein modification and lipid peroxidation in plasma (in vitro) [45].

Resveratrol has also been found to inhibit platelet function, which may be associated with partial modification of flux in the polyphosphoinositide cycle and a decrease in phosphatidylinositol 4,5-bisphosphate available for signaling in blood platelets. Inhibition of polyphosphoinositide metabolism may be associated with inhibition of PI-4(5)-kinase and protein kinase C. In addition, resveratrol has been found to reduce cyclooxygenase activity and inhibit arachidonic acid metabolism in platelets in vitro [39]. Gligorijevic et al. [46] report that resveratrol may bind to fibrinogen in vitro and protect it from harmful oxidation, as indicated by various spectroscopic methods. Other results indicate that resveratrol may interact with thrombin, which is an important protein for platelet activation and the coagulation process; Shahidi et al. [47] found that resveratrol demonstrated both anticoagulant and antifibrinolytic potential in human umbilical vein endothelial cell (HUVECs) cell culture.

Other results indicate that long-term resveratrol supplementation with doses between 100 and 200 mg/day has beneficial effects on major risk factors for stroke, such as weight status, blood pressure, lipid profile and glucose level. For example, Olas and Wachowicz [39] described that resveratrol modulates the signal transduction in blood platelets in different and sometimes in opposite ways. Its antiplatelet action may correlate with partial modification of flux in the polyphoshoinositide cycle. In addition, it causes the changes in reactive oxygen species level. Resveratrol inhibits also the activity of cyclooxygenase and reduces arachidonic acid metabolism.

Although the therapeutic potential of resveratrol in COVID-19-associated hemostatic disorders is also reviewed by Giordo et al. [30], the authors do not present any results of in vitro or in vivo experiments using elements of hemostasis (blood platelets, whole blood or other) treated with resveratrol during COVID-19. They only speculate that resveratrol and its anti-thrombotic and anti-inflammatory properties may serve as a compound for slowing and ameliorating the phenomena associated with severe COVID-19, including vascular thrombosis. They suggest that resveratrol may affect on primary hemostasis, including the suppression of blood platelet activation (especially platelet aggregation) and secondary hemostasis (modulating coagulation process by the inhibition of tissue factor/factor VIIa complex formation). Anti-platelet action of resveratrol probably involves the inhibition of cyclooxygenase activity, lowering nitric oxide concentration, decreasing cytoplasmic Ca^2+^ and blocking Ca^2+^ entry into blood platelets [57]. Results of Jang et al. [58] suggest that the anti-platelet action of resveratrol is associated with the inhibition of NADPH oxidase—derived ROS production and subsequent oxidative inactivation of SH2 domain-containing protein tyrosine phosphatase-2.

Resveratrol analogues have also anti-platelet potential. Pterostilbene (a dimethylether analogue of resveratrol) possesses high potency in the prevention of blood platelet activation in humans. For example, it inhibits platelet aggregation stimulated by collagen and reduces P-selectin exposition [59]. Revishankar et al. [60] also observed that other resveratrol analogue—isorhapontigenin inhibits blood platelet activation stimulated by ADP (possible via P2Y12 receptor).

### 2.2. Curcumin

Another natural phenolic compound with antioxidant properties is curcumin, which has been used in natural medicine for thousands of years. It is a plant compound found in the rhizome extract of *Curcuma longa* L, and is commonly used as a spice known as turmeric. It is used to impart a strong yellow color to foods, most famously Indian curry, but also to a range of other foods, such as margarine, mustard, pasta or soft drinks; it is known to be safe for consumption (E100). In traditional Indian and Chinese medicine, it was used to quell excessive appetite and to treat jaundice and other diseases of the liver and digestive system, as well as colic, tooth and chest pain, and menstrual pain. It was also used as a means of supporting wound healing and treating skin discoloration [61].

In modern natural medicine, curcumin is regarded as an antiseptic, anti-inflammatory, antioxidant, anti-platelet and anti-cancer agent [61,62], and is recommended for treating eye infections, skin conditions such as burns, bites and acne, and diseases of the digestive system such as diarrhea, indigestion, hyperacidity, flatulence and ulcers. It has even been recommended as an antidepressant.

However, curcumin has very low bioavailability and is metabolized quite quickly. For example, studies conducted on rats in which curcumin was administered orally at a very high dose (1 g/kg body weight) found that 75% was excreted in the feces and only trace amounts were found in the urine. More than 60% of the curcumin excreted in the feces were its metabolites. It has however been found that curcumin bioavailability may be increased by various compounds, including piperine, a major active component of black pepper [27].

Curcumin has the capability to scavenge various intracellular small oxidative molecules; it can also up-regulate the expression of glutathione and reduce the production of ROS [27]. Kolodziejczyk et al. [48] describe the antioxidative potential of curcumin (12.5–50 μg/mL) in human blood platelets and plasma treated with ONOO^−^ (in vitro), and Manikandan et al. [49] indicate that curcumin modulates ROS production in myocardial ischemia in rats, probably by inhibition of xanthine dehydrogenase and xanthine oxidase.

Kim et al. [50] report that curcumin and its derivatives demonstrate anticoagulant properties in vitro and in vivo; they appear to prolong prothrombin time and inhibit the generation of thrombin and activity of coagulation factor Xa in human plasma. Moreover, curcumin demonstrated a stronger anticoagulant effect than its derivatives, indicating that the methoxy group in curcumin positively regulated its anticoagulant function.

Madhyastha et al. [63] found curcumin to facilitate fibrinolysis and cellular migration by modulating urokinase plasminogen activator expression (uPA), and that uPA upregulation was dependent on activation of c-Jun N-terminal kinase (JNK) and p38 mitogen-activated protein kinase (MAPK) signal pathways. Interestingly, however, Liu et al. [64] report that oral administration of curcumin at 25, 50 and 100 mg/kg for seven days has no effect on antiplatelet and anticoagulation properties in Wistar rats.

More details about the positive activity of curcumin in hemostasis, including its effects on platelet functions are reviewed by Kaihanian et al. [61] and Tabeshpour et al. [62]. Recently, Rukoyatkina et al. [65] have identified molecular mechanisms involved in the inhibitory effects of curcumin on blood platelet activation. It by activation of adenosine A_2A_ receptor stimulated protein kinase A activation and phosphorylation of vasodilator-stimulated phosphoprotein. Results of Chapman et al. [66] also demonstrated that the major plasma metabolite of curcumin—tetrahydrocurcumin may modulate hemostasis: platelet aggregation and coagulation process. Li et al. [67] suggest that anti-platelet action of tetrahydrocurcumin includes inhibiting MAPKs/phospholipase A_2_ (PLA_2_) signaling, attenuating platelet TXA_2_ generation, and granule secretion.

However, there is a lack of information suggesting that curcumin may have a positive role for treating CVDs in patients with COVID-19. Despite this, some review papers have examined the potential effects of curcumin in the treatment of COVID-19, in particular the value of curcumin supplementation as an anti-inflammatory and antioxidant compound in patients with COVID-19 [68]. Moreover, the potential of effect of curcumin may include inhibiting the entry of virus to the cells, inhibiting encapsulation of the virus and viral protease, and modulating different cellular signaling pathways, for example curcumin has strong inhibitory effect on nuclear factor-kappa B (NF-κB) [68].

### 2.3. Quercetin

One of the most important phenolic compounds in the diet is quercetin (from the Latin *quercetum*—oak forest). It is an almost ubiquitous compound supplied to the body via a fruit and vegetable diet. Quercetin is most often found in the form of glycosides, i.e., sugar derivatives, e.g., rutin, although it can also occur in a free form. It is commonly found in tea, red wine, onions, apples and medicinal herbs, although its highest concentration has been noted in capers (234 mg per 100 g of edible portion). Its daily consumption is estimated at 25–50 mg [69,70,71]. However, quercetin has a low bioavailability after oral administered [69,70,71].

The biological action of quercetin results from its high antioxidant activity and the ability to inhibit enzymes involved in the formation of the inflammatory process. The antioxidant properties of quercetin result from the presence of hydroxyl groups at positions C-3, C-5, C-7, C-3′ and C-4′; a double bond in position C2-C3, as well as a carbonyl group in position C-4. Its antioxidative properties are based on various mechanisms: (I) scavenging ROS, inter alia by inactivating produced oxygen radicals (hydroxyl radical, singlet oxygen and lipid radicals); (II) limiting the production of ROS in cells by inhibiting the activity of enzymes involved in their formation (xanthine oxidase, membrane oxidase NAD(P)H, myeloperoxidase); (III) chelation of transition metal ions (copper and iron), which prevents the formation of reactive hydroxyl radicals in cells, an important factor in the pathogenesis of many diseases; (IV) breaking the cascade of free radical reactions in enzymatic and non-enzymatic lipid peroxidation; and (V) by possibly affecting the transmission of intracellular signals by protein kinases. Quercetin also exhibits significant heart-related benefits, such as inhibition of low density lipoprotein (LDL) oxidation and exerting a protective effect on NO^.^ function under oxidative stress. The compound has also been found to influence the oxidative profile in blood platelets of rats with hypothyroidism [72].

Mosawy et al. [51,52] studied the effect of 6 mg/kg quercetin, administered daily for seven days, on blood platelet aggregation, platelet secretion and vessel occlusion in a mouse model of platelet mediated arterial thrombosis. The flavanol appeared to inhibit the activation of platelets stimulated with protease-activated receptor 4 (PAR4). In addition, quercetin (6 mg/kg, daily for seven days) was found to reduce platelet aggregation stimulated by 250 μM PAR4 in diabetic mice (C57BL/6 mice); in this case, diabetes was confirmed when the blood glucose level was ≥13 mmol/L [53]. Navarro-Nunez et al. [54] indicate that quercetin inhibits platelet adhesion to collagen and fibrinogen, and its action is correlated with inhibition of the activity of various kinases, such as Src kinases. Moreover, quercetin inhibits platelet aggregation stimulated by ADP [55]. However, metabolites of quercetin can be responsible for the antiplatelet effect [73,74]. For example, methycatechol formed by human microflora has a strong antiplatelet effect. Its mechanism of action is mainly based on the effect on intracellular calcium signaling [73].

Choi et al. [56] demonstrated that quercetin and isoquercetin inhibit the enzymatic activity of thrombin and coagulation factor Xa, and suppress fibrin clot formation and blood clotting. The in vivo and ex vivo anticoagulant activities of quercetin and isoquercetin were evaluated in a thrombin-induced acute thromboembolism model and ICR mice. Interestingly, quercetin was found to demonstrate weaker anticoagulant properties than isoquercetin, probably because of the differences in their chemical structure. Further information about the biological importance of quercetin has been provided in a review paper by Anand David et al. [69].

Recently, Derosa et al. [33] have indicated that quercetin appears to have a significant function in COVID-19 treatment. Its effectiveness against COVID-19 may include its effect on SARS-CoV-2 proteases, and angiotensin-converting enzyme 2 (ACE2). As such, more studies are needed of its multifaceted activity.

A summary of the relationship between dietary phenolic compounds with antiviral potential and CVDs, and their possible role in supporting protection against CVDs in patients with COVID-19 is given in Figure 1.

## 3. Conclusions

The antioxidant, anti-platelet and anticoagulant properties of phenolic compounds have been well documented in a number of in vitro and in vivo studies, which have demonstrated they can play a positive role for the prophylaxis and treatment of CVDs. However, it is important to emphasize that their role in protecting against CVDs in patients with COVID-19 remains only theoretical. No credible evidence for their efficiency in humans or animals have been reported, and claims regarding their efficiency do not have any basis even in in vitro studies. Therefore, it can not currently be unequivocally stated whether phenolic compounds can yield benefits in treating CVDs in COVID-19 patients.

## Figures and Tables

**Figure 1 nutrients-14-01390-f001:**
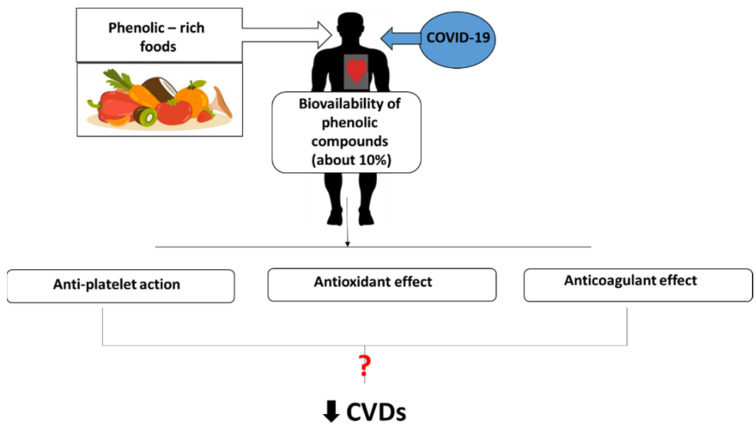
The relationships between dietary phenolic compounds with antiviral potential and CVDs, and their presumed role in protecting against CVDs in patients with COVID-19.

**Table 1 nutrients-14-01390-t001:** Main dietary sources of phenolic compounds with antiviral potential and their effectiveness against modulation of hemostasis and CVDs [24]; (modified). COX—cyclooxygenase; LDL—low density lipoprotein; NO—nitric oxide; ROS—reactive oxygen species.

Phenolic Compound with Antiviral Potential	Main Source	Food Concentration	Effectiveness against Modulation of Hemostasis and CVDs	References
Resveratrol (3,4′,5—trihydroxystilbene)	Grapes, peanuts, pine, mulberry, black currant, strawberries, raspberries, apples	Grapes: 50–400 μg/g fresh weight; Red wine: 0.1–7 mg/L; White wine: 0.04 mg/L; Grape juice: 0.05 mg/L	Antioxidant activity: inhibition of lipid peroxidation, nitration and oxidation of blood platelet and plasma proteins (in vitro); Anti-platelet, anti-coagulant and antifibrinolytic activity: inhibition of ROS production, inhibition of activity of various enzymes, including COX (in vitro and in vivo)	[39,40,41,42,43,44,45,46,47]
Curcumin (1,7-bis-(4-hydroxy-3-metoxyphenylo)-1,6 heptadiene 3,5-dion)	Turmeric	12.5–50 μg/mL (in vitro) 3 000 mg/100 g (in vivo) 0.1–50 μM (in vitro) 100 mg/kg (in vivo)	Antioxidant activity: inhibition of lipid peroxidation, nitration and oxidation of blood platelet and plasma proteins (in vitro and in vivo); Anticoagulant potential: prolongation prothrombin time and inhibition the generation of thrombin and activity of coagulation factor Xa (in vitro and in vivo)	[48,49,50]
Quercetin (3,3′,4′,5,6-pentahydroxyflavone)	Capers, buckwheat, onions, black tea, red wine, apples	Capers (raw) 234 mg/100 g; Buckwheat: 184–535 mg/100 g; Onions: 120 mg/100 g; Black tea: 10–20 mg/g	Antioxidant activity: inhibition of LDL oxidation and the protective effect on NO^.^ function under oxidative stress (in vitro and in vivo), Anti-platelet, anti-coagulant and antifibrinolytic (in vitro and in vivo)	[51,52,53,54,55,56]

## Data Availability

Not applicable.

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
