# Peer review of "The Antioxidant, Anti-Platelet and Anti-Coagulant Properties of Phenolic Compounds, Associated with Modulation of Hemostasis and Cardiovascular Disease, and Their Possible Effect on COVID-19"

_nutrients, 2022, doi:10.3390/nu14071390_

Round 1
Reviewer 1 Report
This study well designed and written. This topic is very important and interesting.
I recommend its pulication as it has valuable informations about the biological and antioxidant properties of phenolic compounds agianst hemostasis and cardiovascular disease, and their possible effect on COVID-19.
Notes:
- I suugest that author should add the number of articles which were choosed for constructing this review after his searching using the different keywords.
- Author should adher strictly to the journal guidlines in wrotting the references.
- Fullstop should be added after each word for the abbrevaiated journal names and also the year should be bold.
Author Response
This study well designed and written. This topic is very important and interesting.
I recommend its pulication as it has valuable informations about the biological and antioxidant properties of phenolic compounds agianst hemostasis and cardiovascular disease, and their possible effect on COVID-19.
I would like to thank the Reviewer for providing helpful comments.
Notes:
- I suugest that author should add the number of articles which were choosed for constructing this review after his searching using the different keywords.
Response: I have added this information: “The following terms were used (publication date – 10 years): “COVID-19” (the number of articles: 201,341), “phenolic compounds” (the number of articles: 45,342), or “phenolics” (the number of articles: 238,876), or “phenolic compound with antiviral activity” (the number of articles: 987),, “oxidative stress” (the number of articles: 156,765), or “antioxidant” (the number of articles: 445,654),, and “hemostasis” (the number of articles: 98,321), or “coagulation” (the number of articles: 300,378), or “blood platelets” (the number of articles: 78,765), or “thrombosis” (the number of articles: 187,987), or “cardiovascular diseases” (the number of articles: 678,654).”
- Author should adher strictly to the journal guidlines in wrotting the references.
Response: I have corrected.
- Fullstop should be added after each word for the abbrevaiated journal names and also the year should be bold.
Response: I have corrected.
Reviewer 2 Report
Thank you for the opportunity to assess this mini-review article entitled "The antioxidant, anti-platelet and anti-coagulant properties of phenolic compounds, associated with modulation of hemostasis and cardiovascular disease, and their possible effect on COVID-19". The topic is of interest, within the scope of the journal. However, according to the author guidelines, I am unsure that a mini-review format is acceptable.
Major comments
- The introduction occupies a high proportion of the total manuscript. I suggest reducing its length by incorporating some of its information into other manuscript sections.
- line 53: It could be the antioxidant and anti-inflammatory properties that may improve COVID-19 prognosis. It would also be more helpful to cite some original research of antiviral properties instead of reviews (examples: 10.1016/j.ejphar.2021.174615, 10.1002/ptr.6916, 10.1128/AAC.00772-21, 10.3390/v13071335, 10.1016/j.foodchem.2021.131594)
- lines 92-97: A small section entitled literature search strategy could include the information.
- lines 153-154: Since the review also deals with the antiplatelet action, it is advisable that all of the involved properties are described, not just mention that these were reviewed in another article.
- line 164-166: also discuss antioxidant mechanisms (example: (10.1016/j.freeradbiomed.2015.10.413) and recent studies of resveratrol analogues (examples: 10.1021/acs.jafc.1c00367, 10.1016/j.ejphar.2019.172627). Also provide citations for the so-far mentioned mechanisms (for example calcium handling: 10.1186/s12199-020-00905-1)
- line 206-207: Please expand on the hemostatic and platelet effects of curcumin, instead of just referencing the reviews. The topic of this review needs to be adequately covered. Also novel studies could be included (examples: 10.1016/j.thromres.2019.04.029, 10.1055/s-0041-1735192)
Minor comments
- line 33: infection with SARS-CoV-2 (virus), not COVID-19 (disease)
- line 41: 20 references are used for the importance of oxidative stress in COVID-19, this is inappropriate.
- Table 1: add footnotes with the abbreviations, also change the order of the compounds according to the order of their corresponding sections.
- line 107: it should be section 2.1.
- line 161: may affect (not effect on) primary hemostasis
- line 178-180: please provide a reference
- line 182-186: please provide a reference
- I would suggest Figure 1 is placed somewhere within the text, instead of the conclusion.
- The reference list contains twice the number of the citation, please correct.
Author Response
Thank you for the opportunity to assess this mini-review article entitled "The antioxidant, anti-platelet and anti-coagulant properties of phenolic compounds, associated with modulation of hemostasis and cardiovascular disease, and their possible effect on COVID-19". The topic is of interest, within the scope of the journal. However, according to the author guidelines, I am unsure that a mini-review format is acceptable.
I would like to thank the Reviewer for providing helpful comments.
Major comments
- The introduction occupies a high proportion of the total manuscript. I suggest reducing its length by incorporating some of its information into other manuscript sections.
Response: I have corrected the chapter of Introduction.
- line 53: It could be the antioxidant and anti-inflammatory properties that may improve COVID-19 prognosis. It would also be more helpful to cite some original research of antiviral properties instead of reviews (examples: 10.1016/j.ejphar.2021.174615, 10.1002/ptr.6916, 10.1128/AAC.00772-21, 10.3390/v13071335, 10.1016/j.foodchem.2021.131594)
Response: I have added new papers:
Yang, M., Wei, J., Huang, T., Lei, L., Shen, Ch., Lai, J., Yang, M., Liu, L., Yang, Y., Lin, G., Liu, Y. Resveratrol inhibits the repoli-cation of severe acute respiratory syndrome coronavirus 2 (SARS-CoV-2) in cultured Vero cells. Phytother. Res. 2021, 3, 1127-1129.
Tietjen, Y., Cassel, J., Register, E.T., Zhou, X.Y., Messick, T.E., Keeney, F., Lu, L.D., Beattie, K.D., Rali, T., Tebas, P., Ertl, H.C.J., Salvino, J.M., Davis, R.A., Montaner, L.J. The natural stilbenoid (-)-hopeaphenol inhibits cellular entry of SARS-CoV-2 USA-WA1/2020, B.1.1.7, and B.1.351 variants. Antimicrob. Agents Chemother. 2021, 65, 1-20.
ter Ellen, B.M., Kumar, N.D., Bouma, E.M., Troost, B., van de Pol, D.P.I., van der Ende-Metselaar, H.H., Apperloo, L., van Gosliga, D., van den Berge, M., Nawijn, M.C., van der Voort, P.H.J., Moser, J., Moser, J., Rodenhuis-Zybert, J.A., Smit, J.M. Resveratrol and pterostilbene inhibit SARS-CoV-2 replication in air-liquid interface cultured human primary bronchial epithelial cells. Viruses 2021, 13, 1-17.
Bahun, M., Jukic, M., Oblak, D., Kranjc, L., Bajc, G., Butala, M., Bozovicar, K., Bratkovic, T., Podlipnik, C., Ulrih, N.P. Inhibition of the SARS-CoV-2 3CL pro main protease by plant polyphenols. Food Chem. 2022, 373, 1-11.
- lines 92-97: A small section entitled literature search strategy could include the information.
Response: I have added this section: “This review is based on studies identified in electronic databases, including PubMed, ScienceDirect, Web of Knowledge, Google Scholar, and SCOPUS. The last search was run on December 10, 2021. The following terms were used (publication date – 10 years): “COVID-19” (the number of articles: 201,341), “phenolic compounds” (the number of arti-cles: 45,342), or “phenolics” (the number of articles: 238,876), or “phenolic compound with antiviral activity” (the number of articles: 987),, “oxidative stress” (the number of ar-ticles: 156,765), or “antioxidant” (the number of articles: 445,654),, and “hemostasis” (the number of articles: 98,321), or “coagulation” (the number of articles: 300,378), or “blood platelets” (the number of articles: 78,765), or “thrombosis” (the number of articles: 187,987), or “cardiovascular diseases” (the number of articles: 678,654).”
- lines 153-154: Since the review also deals with the antiplatelet action, it is advisable that all of the involved properties are described, not just mention that these were reviewed in another article.
Response: I have added more information about it: “For example, Olas and Wachowicz (35) described that resveratrol modulates the signal transduction in blood platelets in different and sometimes in opposite ways. Its antiplatelet action may correlate with partial modification of flux in the polyphoshoinositide cycle. In addition, it causes the changes in reactive oxygen species level. Resveratrol inhibits also the activity of COX and reduces arachidonic acid metabolism.”
- line 164-166: also discuss antioxidant mechanisms (example: (10.1016/j.freeradbiomed.2015.10.413) and recent studies of resveratrol analogues (examples: 10.1021/acs.jafc.1c00367, 10.1016/j.ejphar.2019.172627). Also provide citations for the so-far mentioned mechanisms (for example calcium handling: 10.1186/s12199-020-00905-1)
Response: I have added new information - papers:
Marumo, M., Ekawa, K., Wakabayashi, I. Resveratrol inhibits Ca2+ signal and aggregation of platelets. Environ. Health Prev. Med. 2020, 25, 1-8.
Jang, J.Y., Min, J.H., Wang, S.B., Chae, Y.H., Baek, J.Y., Kim, M., Ryu, J-S., Chang, T-S. Resveratrol inhibits collagen-induced platelet stimulation through suppressing NADPH oxidase and oxidative inactivation of SH2 domain-containing protein tyrosine phosphatase-2. Free Radic. Biol. Med. 2015, 89, 842-851.
Huang, W-Ch., Liu, J-Ch., Hsia, Ch-W., Fong, T-H., Hsia, Ch-H., Tran, O-T., Velusamy, M., Yang, Ch-H., Sheu, J-R. Ptherostilbene, a dimethylether analogue of resveratrol, possess high potency in the prevention of platelet activation in humans and the reduction of vascular thrombosis in mice. J. Agric. Food Chem. 2021, 69, 4697-4707.
Ravishankar, D., Albadawi, D.A.I., Chaggar, V., patra, P.H., Williams, H.F., Salamah, M., Vaiyapuri, R., dash, P.R., Patel, K., Watson, K.A., Vaiyapuri, S. Isorhapontigenin, a resveratrol analogue selectively inhibits ADP-stimulated platelet activation. Eur. J. Pharmacol. 2019, 862, 1-13.
- line 206-207: Please expand on the hemostatic and platelet effects of curcumin, instead of just referencing the reviews. The topic of this review needs to be adequately covered. Also novel studies could be included (examples: 10.1016/j.thromres.2019.04.029, 10.1055/s-0041-1735192)
Response: I have added new information - papers:
Rukoyatkina, N., Shpakova, V., Bogoutdinova, A., Kharazova, A., Mindukshev, I., Gambaryan, S. Curcumin by activation of adenosine A2A receptor stimulates protein kinase a and potentiates inhibitory effect of cangrelor on platelets. Biochem. Bio-phys. Res. Commun. 2022, 586, 20-26.
Chapman, K., Scorgie, F.E., Ariyarajah, A., Stephens, T., Enjeti, A.K., Lincz, L.F. The effect of tetrahydrocurcumin compared to curcuminoids on human platelet aggregation and blood coagulation in vitro. Thromb. Res. 2019, 179, 28-30.
Li, W., Ma, J., Zhang, Ch., Chen, B., Zhang, X., Yu, X., Shui, H., He, Q., Ua, F. Tetrahydrocurcumin downregulates MAPKs/cPLA2 signaling and attenuates platelet thromboxane A2 generation, granule secretion, and thrombus growth. Thromb. Haemost. 2021, 1, 1-7.
Minor comments
- line 33: infection with SARS-CoV-2 (virus), not COVID-19 (disease)
Response: I have corrected. Now, it is “Infection with SARS-CoV-2 is known to entail a range of sequelae.”
- line 41: 20 references are used for the importance of oxidative stress in COVID-19, this is inappropriate.
Response: I have corrected. Now, it is: “[26-34]”.
- Table 1: add footnotes with the abbreviations, also change the order of the compounds according to the order of their corresponding sections.
Response: I have added the abbreviations: “COX – cyclooxygenase; LDL – low density lipoprotein; NO – nitric oxide; ROS - reactive oxygen species”. In addition, I have corrected Tables 1.
- line 107: it should be section 2.1.
Response: I have corrected. Now, it is “2.1”.
- line 161: may affect (not effect on) primary hemostasis
Response: I have corrected. Now, it is “may affect on primary hemostasis”.
- line 178-180: please provide a reference
Response: I have added.
- line 182-186: please provide a reference
Response: I have added.
- I would suggest Figure 1 is placed somewhere within the text, instead of the conclusion.
Response: I have corrected. Now, Figure 1 is after chapter - 2.3”.
- The reference list contains twice the number of the citation, please correct.
Response: I have corrected.
Round 2
Reviewer 2 Report
The author adequately addressed my comments.
Author Response
I would like to thank the Reviewer for providing helpful comments.